

# Fluid Conduits and Shallow Reservoir Structure Defined by Geoelectrical Tomography at the Nirano Salse (Italy)

Gerardo Romano[1], Marco Antonellini[2], Domenico Patella[1], Agata Siniscalchi[1], Andrea Tallarico[1], Simona Tripaldi[1], Antonello Piombo[3]

[1]Dipartimento di Scienze della Terra e Geoambientali, Università degli Studi di Bari Aldo Moro, Campus Universitario, via Orabona n. 4, 70125 Bari, Italy
[2]Dipartimento di Scienze Biologiche, Geologiche e Ambientali, ALMA MATER STUDIORUM – Università di Bologna, Piazza di Porta S.Donato 1, Bologna, Italy
[3]Dipartimento di Fisica e Astronomia "Augusto Righi", ALMA MATER STUDIORUM – Università di Bologna, Viale Berti Pichat 6/2, Bologna, Italy

*Correspondence to*: Agata Siniscalchi (agata.siniscalchi@uniba.it)

**Abstract.** Mud volcanoes are fluid escape structures allowing for surface venting of hydrocarbons (mostly gas but also liquid condensates and oils) and water/sediment slurries. For a better understanding of Mud volcanoes dynamics, the characterization of the fluid dynamics within mud volcanoes conduits, the presence, extent, and depth of the fluid reservoirs as well as the connection among aquifers, conduits, and mud reservoirs play a key role. To this aim, we performed a geoelectrical survey in the *Regional Nature Reserve of the Nirano Salse,* located at the edge of the Northern Apennines (Fiorano Modenese, Italy), an area characterized by several active mud fluid vents. This study, for the first time, images the resistivity structure of the subsoil along two perpendicular cross sections down to a depth of 250 m. The electrical models show a clear difference between the Northern and Southern sectors of the area, where the last hosts the main discontinuities. Shallow reservoirs, where fluids muds accumulate are spatially associate to the main fault/fracture controlling the migration routes associated with surface venting and converge at depth towards a common clayey horizon. There is no evidence of a shallow mud caldera below the *Nirano* area. These findings represent a step forward in the comprehension of the *Nirano Salse* plumbing system and in pinpointing local site hazards, which promotes safer tourist access to the area along restricted routes.

## 1 Introduction

Mud volcanoes are fluid escape structures allowing for surface venting of hydrocarbons (mostly gas but also liquid condensates and oils) and water/sediment slurries. Their shape varies in size over different orders of magnitude from centimeters to several hundreds of meters (Manga & Wang, 2015). Mud volcanoes are associated with "mud volcanism" processes (Martinelli & Judd, 2004) that may be quite different ranging from liquefaction due to sediment overpressure and compaction to leakage



from poorly sealed hydrocarbon reservoirs (Kopf, 2002). Worldwide, they are broadly distributed on land and on the sea bottom (Milkov, 2005; Mazzini & Etiope, 2017), especially in contractional areas and fast subsiding basins. Most mud volcanoes are cold seeps, but some are also present in geothermal areas (Amici et al., 2013). Mud transport mechanisms are a

matter of debate being associated both with mud-dyke-sill complexes or with diapirs (Tingay, 2009; Roberts, 2011).

Mud volcanoes are usually associated with quiet and continuous eruptions (Tingay, 2009) but sometimes they explode in dangerous and disruptive events. They are hazardous phenomena, because as of today, it is impossible to define the parameters for modelling the recurrency intervals of the extreme events (Gattuso et al., 2021). Azerbaijan hosts some mud volcanoes displaying the most violent eruptions worldwide; the Lobkatan mud volcano, for example, explodes every 3-5 years in

spectacular eruptions, which send detritus and breccia rafts into the Caspian Sea (Mazzini et al., 2021; Wang & Manga, 2021). In recent times (2006-2016), the massive eruption of the Indonesian Lusi mud volcano, triggered by the drilling of a borehole, continued for years with peak mud flows of 180000 m3/day that caused burial of nearby villages and the displacement of 60000 people (Mazzini et al., 2007; Tingay, 2015). Some historically recorded explosive events occurred at the Salsa di Montegibbio (Sassuolo, Italy), the largest mud volcano in Italy (Borgatti et al., 2019), were reported by Plinius (Historiae

Naturalis) in roman times (50 a.d.) and later (1592-1835) by Biasutti (1907), Govi (1906) and Stöhr (1867). Recently (2014), a sudden massive expulsion of fluids at the Macalube di Aragona mud volcano (Agrigento, Italy) caused the death of two children (Gattuso et al., 2021). Mud volcanoes, therefore, represent potential geohazards, especially where these features are located next to populated areas or where they are tourist destinations.

Mud volcanoes are natural hydrocarbon seeps and as such are important indicators in hydrocarbon exploration where they give

hints about the level of hydrocarbon maturity and the presence of structural highs related to mud diapirism (Warren et al., 2010), fault traps, and their spill-points (Milkov, 2005; Mazzini & Etiope, 2017). Large mud volcanoes provinces are also associated with giant hydrocarbon accumulations and many onshore fields in Europe, Caribbean, Caspian Basin, and Caucasus were discovered by drilling on natural seeps indication (Mazzini & Etiope, 2017). In Azerbaijan, mud volcanoes are associated with structural traps and their feeders are rooted in or below the reservoirs forming an interconnected plumbing system (Planke

et al., 2003). The relationships between production from reservoirs and activity in nearby mud volcanoes is also an issue of interest; usually, the decrease in fluid pressure during exploitation causes diminishing in mud volcanism processes but, in some cases, there is no effect whatsoever (Mazzini & Etiope, 2017).

Mud volcanoes have also important meaning in different historical and societal contexts (Giambastiani et al., 2022). Geo-tourists interested in visiting mud volcanoes are numerous throughout the world (Italy, Colombia, Trinidad, Azerbaijan,

Georgia, Turkmenistan, and Indonesia) for the scenery or for recreational mud bathing due to the beneficial properties of mud for the skin. Mud volcanoes have also a religious significance for the Hindus and for the Temple of Fire worshippers (Ateshgah, near Baku in Azerbaijan), and they were revered at different times by Zoroastrians, Hindus, and Sikhs (Gamkrelidze et al., 2021). The combination of human activities such as geo-tourism, religious worshipping, proximity settlements and mud volcanoes may lead to situations of risk where the fluid emissions are a geohazard.



In the context of mud volcanoes, fluid emissions are geohazards mostly because they can release violently large amounts of
       mud and hydrocarbons and may degrade the soil causing caving and quick-sand effects. Furthermore, they might dissociate
       gas hydrates in submarine environments (Mazzini & Etiope, 2017). Explosive eruption of mud volcanoes with methane self-
       igniting phenomena in Azebaijan are rather common. Tall mud columns ejections up to several and tens of meters have been
       observed during the already mentioned eruption at the Macalube di Aragona as well as in the Trinidad Piparo mud volcano
causing serious risks for the people in their proximity (Mazzini & Etiope, 2017). Mud pools with diameters in the order of a
       few meters, not-existent rims, and depth that can reach 15 m (Giambastiani et al., 2022) may potentially be deadly sinks for
       persons and animals. It is therefore necessary to characterize the fluid dynamics within mud volcanoes conduits, the presence,
       extent, and depth of the fluid reservoirs as well as the connection among aquifers, conduits, and mud reservoirs. Given that it
       is difficult to image fluid sources and conduits below a sedimentary cover, geophysical investigations have the potential to
greatly contribute to this characterization.

       Electromagnetics methods are useful where there is contrast in apparent resistivity due to the presence of mud volcano fluids.
       Transient Electromagnetic (TEM) and the radiomagnetotelluric methods (RMT) were applied by Adrian et al. (2015) at mud
       volcanoes in Perikishkul (Azerabaijan) who observed resistivity variations at shallow (< 10 m with RMT) and intermediate
       depths (150 m with TEM) directly below the surface emissions. Three-D electrical resistivity imaging of mud volcanoes in
New Zealand by Zeyen et al. (2011) shows pipe like structures connected to a deep reservoir; the pipe structures occur along
       a major strike-slip fault plane, which controls fluids surface venting. Geoelectric methods are often combined with other
       geophysical and geological investigations. A multidisciplinary approach of shallow seismic, georesistivity, and
       hydrogeochemical surveys was employed (Rainone et al., 2015) near Pineto (Central Italy). The survey results show that the
       mud reservoirs are not just below the mud volcano, but they are in a fractured zone at 60 m distance from the vent suggesting
the presence of a high-permeability connection (fracture zone) just below the surface clay deposits.

       Several seismic surveys above mud volcanoes have been performed and reported in the literature. Albarello et al. (2012)
       estimated methane emissions at mud volcanoes in Azerbaijan by measuring the seismic tremor at the surface and identified
       energy bursts that could be related to bubbling at depth. Evans (2007) using industry acquired seismic reflection data, well
       data, and satellite imagery characterized the geophysical expression of large mud volcanoes (diameter > 500m) in the South
Caspian Basin and was able to identify a sequence of wedge-shaped layers of erupted sediments with regularly interlayered
       bedding, which could be useful to characterize the temporal time history and recurrency periods of the mud eruptions. Kirkham
       (Kirkham, 2016) also used 3D seismic to characterize mud volcanoes geometry and seismic character, timing and distribution,
       source regions and depletion zones, and understand the mechanisms behind the formation of conduits and ultimately the
       geometry of extruded bodies in the Nile Delta. This latter study was particularly effective in identifying seal bypass conduits
through which large quantities of mud were extruded to the surface.

       Mauri et al. (2018) employing on-land gravimetric surveys at the Lusi mud volcano in Indonesia have shown that this
       geophysical methodology is effective in characterizing sediment density variations within the basin and in the mud edifice,
       which can then be related to the presence of fault and fracture systems. A three-dimensional deep electrical tomography survey



was also performed at the Lusi mud volcano in Indonesia, by Mazzini et al., (2021) who point out the presence of a fault and fracture system just underneath the volcano (Mauri et al., 2018) and image a region where a mix of groundwater, mud breccia, hydrocarbons, and boiling hydrothermal fluids are stored below the subsided area in proximity of the major vent.

Geodetic surveys are, also important for mud volcanism characterization. Satellite Differential Interferometry Synthetic Aperture Radar (DInSAR) has been useful for detecting pre-eruptive ground deformation in Azerbaijan mud volcanoes by Antonielli et al. (2014) who have shown that inflation (vertical uplift up to 20 cm) and subsidence could be related to the mud
flowing in shallow reservoirs.

The objective of our work is to use deep electric geoelectric tomography surveys at the Nirano Salse in Italy to characterize the fluid dynamics within mud volcanoes conduits, the presence, extent, and depth of the fluid reservoirs as well as the connection among aquifers, conduits, and mud reservoirs. This is necessary to assess the risk level due to mud venting geohazards in this natural reserve, which every year is visited by tens of thousands of school students and tourists. This study,
for the first time, presents a clear image of the subsoil along two perpendicular cross sections down to a depth of 250 m imaging structures of which nothing was known before. Furthermore, this geophysical investigation well integrates with already existing geophysics in the area such as that of Accaino et al. (2007) based on 3D seismic data and 2D geoelectric tomography and of Antunes et al. (2022) who used P wave analysis of drumbeat signals and vertical to horizontal seismic amplitude to detect mud movement in the subsoil.

Our study, by imaging low resistivity subsurface aquifers/reservoirs where high salinity fluids muds concentrate and the fault/fracture-controlled migration routes associated with surface venting, represent a step forward in the comprehension of the Nirano Salse plumbing system previously only inferred, based on hydrogeologic data, by Giambastiani et al. (2022).

**2 Geological introduction to the area**

The Regional Nature Reserve of the *Nirano Salse* (*Nirano Salse* from now on) is located at the edge of the Northern Apennines
(Fiorano Modenese, Italy), an area that is characterized by several active mud fluid vents.

The *Nirano Salse*, known since the Roman Times, are in a beautiful and scenic landscape of Northern Italy. With a surface of approximately 75,000 m2, *the Nirano Salse* is one of the largest mud volcano areas of Italy and Europe, as well as one of the most visited (more than 50000 visitors each year).

The *Nirano Salse* are located within a geomorphological bowl (Fig. 1), which resembles a collapsed caldera. The diameter of
the bowl is about 800 m and the difference in elevation between the bowl's rim and its bottom is about 50-60 m. The structure is almost completely circular except in the SE corner where a small creek cuts across the rim and two relatively NW-SE oriented steep ridges border the southwestern flank of the circular structure. Mud pools and gryphons are located within the caldera-like structure in three major groups two in the northern part and one in the southern part of the bowl.

There are several individual active cones, gryphons, and subcircular pools distributed at the base of the caldera within an area
of recently extruded mud that is free of vegetation. The specific number and location of the cones is rather constant over time



except some small local changes in vents activity. Volcano morphology at the site depends mostly on the characteristics of extrusion style (abrupt Strombolian-type eruptions forming gryphons), persistence of degassing produced by the rising to the surface of salty and muddy water mixed with gas (mostly CH4), and connection with groundwater aquifers (mud pools) (Giambastiani et al., 2022).

The Nirano Salse fluid venting occurs in the Plio-Pleistocene Argille Azzurre Fm (FAA), which in the south-eastern part of the area are just a thin veneer of transgressive clays (less than 50 m) above the Tortonian Termina Fm, which includes clay breccia, marls, and sandstone (0-600 m in thickness). Below the Temina Fm, the turbiditic well-layered Burdigalian-Langhian Pantano Fm (about 200 m in thickness) represents the maximum limit to which the geoelectric survey can extend to. The stratigraphic sequence below the FAA is deformed by two systems of high angle faults; one NW-SE oriented with vertical

offsets and one oriented NE-SW with apparent strike-slip offsets (mostly left-lateral) (Gasperi et al., 2005).

The Nirano mud volcanoes have been interpreted so far as to be just above a NW-SE blind thrust anticline (Bonini, 2008; Bonini, 2009; Bonini, 2012). The gas emissions would be the surface expression of fluids escaping from a deep leaky reservoir (about 1.5 km depth) located in in permeable Epiligurian units of Eocene-Miocene Age (Bonini, 2007; Bonini, 2008). Any gas accumulation within a subsurface reservoir would increase the pore pressure due to buoyancy (Dasgupta & Mukherjee, 2020);

if the sealing unit is of poor quality, the gas may rise towards the surface either following faults and fracture zones or fingering through poorly consolidated mud sediments.

### 3 Geophysical investigations in Nirano Salse area

### 3.1 Previous geophysical data

Several geophysical studies were conducted in the last decades to characterize the *Nirano Salse* subsoil and to address some

of the open issues regarding the fluid origins in the area. Most of them are small scale and relatively high-resolution studies, focused on the determination of the shallow (from the surface down to 50 m below the surface) subsoil structure for the identification of the superficial outlet of the volcanic conduits and chimneys and possible fluid reservoir. Accaino et al. (2007) performed tomographic inversion of first arrivals of 3D seismic data, and 2D geo-electrical surveys on the South-Westernmost mud volcano (here named MV1). Focusing on the geoelectrical data, the overall retrieved electrical properties distribution was

characterized by low resistivity values (from 3 to 5 Ωm) associated with fluid-saturated sediments occurring in the mud volcano area and higher resistivities (up to 40 Ωm) observed in the pelitic sediments surrounding the mud volcano apparatus. Moreover, the used interelectrodic spacing, equal to 3 m, was enough to provide a resolution sufficient to identify the sub-vertical structures of the superficial outlet of the volcanic conduits and chimneys and a mud chamber at a depth of 25 m. Similar results were obtained by Lupi et al., (2016) who presented a more extended geoelectrical survey. In this latter study, a longitudinal

geoelectrical profile, passing near all principal mud volcanoes in the area and with an interelectrodic spacing of 5 m, depicted dome-like reservoirs at about 20 m below the ground surface (ca. 50 m wide and more than 20 m thick) feeding the mud vents. The Authors supposed that the fluid transfer from the fluid reservoirs to the vents at the surface may occur along narrow



conduits that are beyond the resolution limits of the conducted ERT survey they performed. The conductive domes positively correlate with the presence of high permeability areas that act as preferential leakage pathways for gas migration as observed by Sciarra et al. (2019). Finally, Giambastiani et al. (2022) assumed that these permeable areas are none other than shallow aquifers with variable size and thickness possibly leaking to the surface along faults formed during the subsidence of the area and the formation of the caldera-like morphology (bowl).

To the Authors knowledge, no other geophysical data acquired or describing the Nirano Salse area are available or published at the present time.

## 3.2 New geophysical data

To gather information both on the lateral and in-depth structure of the Nirano Salse area, and to investigate a possible deep connection between the conductive domes found by Lupi et al. (2016), two ERT profiles (ERT_N1 and ERT_N2 in what follows) were acquired along orthogonal transects passing by the mud volcanoes area and extending from "rim" to "rim" of the caldera-like structure (figure 1). The length of the profiles was conditioned by the harsh topography over the "rims" which limited the profile extension to 820 m with an interelectrodic spacing of 20 m and a total of 42 electrodes deployed. Profile ERT_N1 trace overlaps Lupi et al. (2016), ERT survey to allow a direct comparison between the previously acquired geoelectrical data and the one that we present here. Along each of the profiles, data were acquired in Wenner-Schlumberger (WS), Dipole-Dipole (both in direct and reverse configurations) (DD and DD-rev) and Pole-Dipole (PD) by using a Syscal Pro 48 ch (Iris instruments). To perform pole-dipole surveys, a remote pole was placed ~2.5 km away from the crossing point of the ERT profiles. Among standard arrays, DD is the most advocated for the study of lateral resistivity contrast, but it is also the one with the lower S/N ratio, WS should provide a better vertical resolution and a good S/N ratio while P-DD has an intermediate lateral and vertical resolution, a good S/N ratio, and a deeper investigation depth. For a comprehensive review of the arrays' characteristics in terms of resolution, investigation depth and sensitivity to noise see Dahlin & Zhou, (2004), or Martorana et al. (2017).





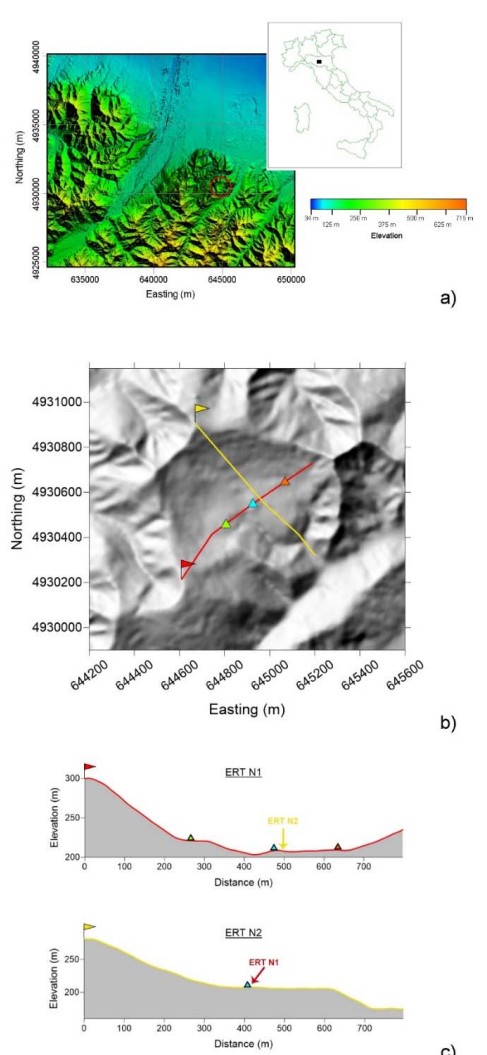


**Figure 1: upper panel: location of the Nirano Salse area. Central panel: ERT profiles locations on top of a DEM. The red line indicates the position of ERT N1 profile. The yellow line indicates the position of ERT N2 profile. The red and the yellow flags indicate the location of the first electrode on both profiles. Green, cyan and orange triangles indicate, respectively, the location of MV1, MV2 and MV3. Lower panels: Elevation profiles corresponding to ERT N1 and ERT N2 profile traces.**



Electrode-ground contact resistance values were checked before measurement and, where necessary, lowered by using salty water. During acquisition, these values were stored to better identify noisy/bad data presence. In fact, as reported in Zhou & Dahlin, (2003), the contact resistance values storage may be fundamental in evaluating whenever or not anomalous data must be considered outliers or not. ERT error outliers, in fact, are often correlated with high contact resistances for some of the electrodes used in a measurement. Moreover, the influence of the measurement errors on the geoelectric data inversion will

also be accounted for by integrating the errors in the inversion procedure.

Measurement errors can be derived mainly from the stacking procedure, from the repetition of the ERT measurement sequence or by comparing the direct measurements with the reciprocal ones. The stacking procedure is the simplest method of assessing a measurement error in a ERT and it consists in repeated measurements of the transfer resistance through several cycles of current injection. The repeatability of an ERT consists in performing the same complete ERT survey to obtain multiple and

independent measurements of the transfer resistance from which to derive error estimates. The comparison of the direct measurements with the reciprocal ones is based on the reciprocity principle which states that the switching of source/sink and observation locations would yield the same response (Parasnis, 1988). In practise, reciprocity checks for ERT are conducted by swapping the current and potential electrodes. A detailed comparison of stacking, repeatability, and reciprocal errors and of their utility to describe errors in measurements can be found in Tso et al. (2017). According to Tso et al. (2017), stacking

errors are potentially an inadequate measure for describing the true quality of ERT measurements since they are generally smaller than repeatability and reciprocal errors. Although stacking errors are the less reliable, due to survey time optimization, they are those mostly considered in standard ERT applications; reciprocal errors are often calculated for Dipole-Dipole measurements and repeatability is generally applied only in time lapse experiments.

With the aim to maximize all available information, we will first focus on the ERT_N1 geoelectrical profile where a

comparison with the resistivity model obtained by Lupi et al., (2016) is also possible. As for our dataset, since reciprocal measurements and relative estimates are available only for DD data, for the other configurations, only stacking errors will be considered. Therefore, we firstly analyse the Dipole-Dipole dataset comparing stacking and reciprocal error distributions as well as the results obtained by the inverse modelling using both type of error estimates. Successively, the reliability of all the available models will be evaluated by verifying their consistency.


### 3.2.1 ERT_N1 profile

#### 3.2.1.1 Dipole-Dipole data inversion

The starting dataset was composed by 851 measurement points. The contact resistance values result all good with values well below 2 KΩ. By considering the whole datasets, without filtering out any measurements, the mean contact resistance was 0.37

KΩ ± 1.54 KΩ for the direct DD measurement and 0.17 KΩ ± 0.35 KΩ for the reverse DD. The higher standard deviation associated with the direct DD measurement can be ascribed to a not perfect contact of an electrode placed on a paved road



(intercepting ERT N1 profile at about 700 m, contact resistance equal to 28 KΩ), which was lately improved by adding salty water. Hence, it can be excluded that anomalous or negative resistivity data are associated to bad contact resistances.

As expected, and known from literature, stacking errors, calculated on direct DD measurements, are generally lower than
reciprocal errors (figure 2) (Tso et al., 2017). In the central portion of the experimental pseudosection, the different behaviour of the error spatial distributions is striking with the reciprocal errors three times larger than the stacking errors. By limiting the colour scale range of the stacking error map from 0% to 10% (not shown here), this last error spatial distribution clearly shows the reversed "v" shape (figure 2a). This is a proof that the stacking errors tend to underestimate the effective measurement errors. However, it may also indicate that the two different kinds of errors follow the same general spatial distribution. When
reciprocal or repeatability errors are not available, a cautious use of the stacking errors could be done by multiplying them by a constant factor. The effect of increasing the electrode spacing on data quality is also apparent in the distribution of the percentage stacking errors. The deeper data show, on average, higher percentage stacking errors.

Measurements with an associated percentage reciprocal error larger than 40% were discarded from the original datasets. Most of the discarded data were localized in the surficial part of the pseudosection, between 300 m and 500 m along the profile. An
additional filtering stage was based on the observation of resistivity data as a profile plot. Data characterized by abrupt resistivity variations and negative apparent resistivity values, probably due to the existence of large resistivity contrasts (Lee & Cho, 2020), were also filtered out. In total, almost 10% of the original data was excluded during the filtering stages.



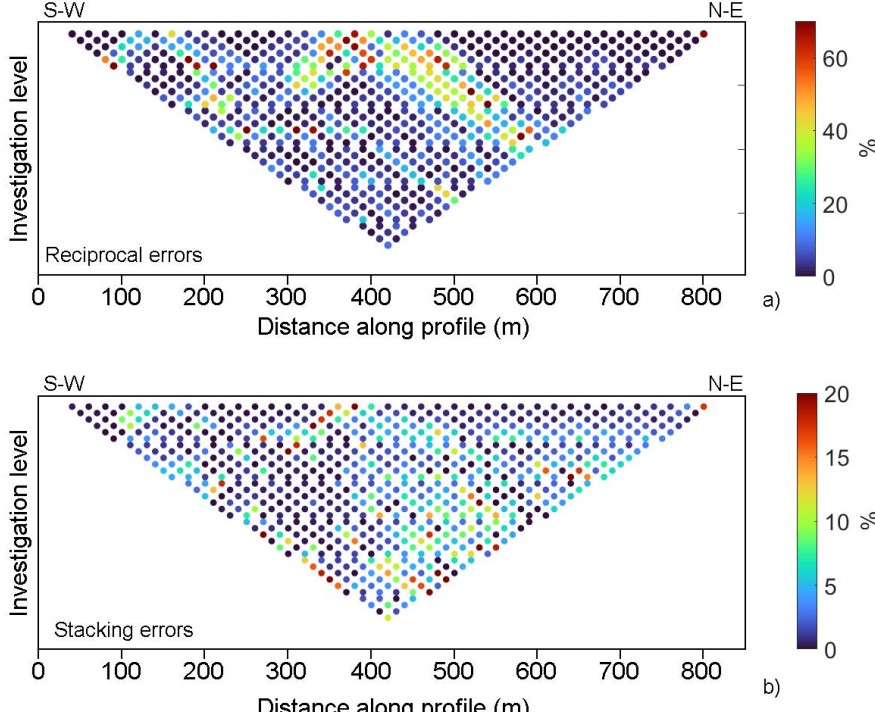

**Figure 2: Panel a): distribution of the reciprocal percentage errors. Panel b): distribution of the percentage stacking errors**
**calculated on DD direct measurements. The error magnitude is given by the colour scale placed on the right side of each panel. Note**
**that the limits of the two-colour scales are different.**

Finally, topographical information was included into the dataset which was inverted with RES2DINV program (Geotomo
Software, Loke & Barker, 1996). The resistivity models presented in what follows were obtained by using a L2 norm and
directly inverting the apparent resistivity values that better converged.

The two models presented in figure 3 show the inversion results obtained including in the inversion procedure both the
reciprocal errors (fig. 3, panel a) and the stacking errors (fig. 3, panel b). The use of reciprocal errors in the inversion procedure
leads to a model with a low RMS value (4.8%) and with resistivity contrasts smoother than those visible in the model with the
staking errors which, being smaller than the reciprocal ones, also influenced the final RMS errors (9.9%). Generally, the models
show a similar pattern of resistivity distribution within the subsoil where:
- the SW part of the section is more resistive than the NE one



- a highly conductive surficial anomaly placed between 300 and 400 meters (horizontal distance) in the area less constrained by the data due to filtering operations
- a shallow highly conductive zone below the mud volcanoes, which is characterized by a lateral variability
- an intermediate relatively resistive layer, which appears to be more continuous in the model obtained by integrating the stacking errors
- two deep conductive zones placed on the SW and NE bottoms of the sections

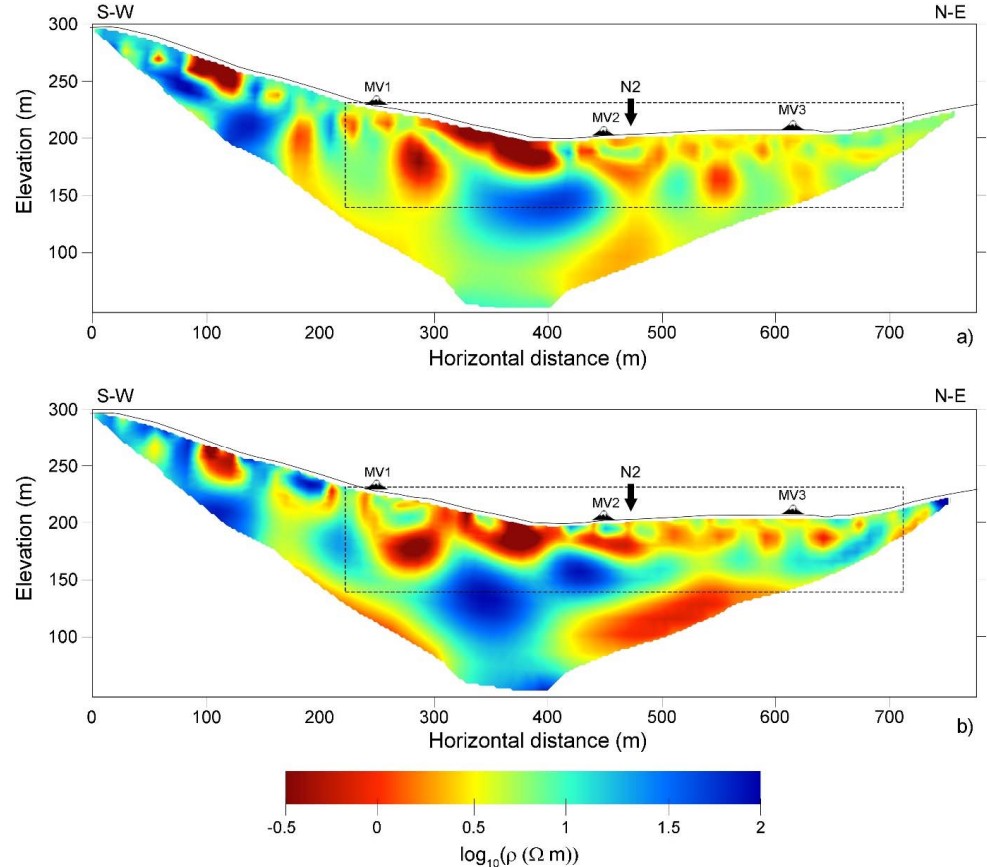

**Figure 3: resistivity models obtained integrating error estimates: panel a) reciprocal errors; panel b) stacking errors. The two model**
**are presented with the same colour scale. In the sections, it is also reported the location of the mud volcanoes (MV1, MV2 and MV3),**



**the crossing point between the ERT_N1 geoelectrical profile and the ERT_N2 ones (black arrow with the N2 label) and the subsoil portion investigated by Lupi et al. (2016) (identified by a black dashed line).**

Considering the similarity between the two models and assuming a higher reliability level for the one obtained by integrating
the reciprocal errors, a comparison with Lupi et al. (2016) results is presented in figure 4. The colour scales adopted are similar although not the same; the differences, in any case, are not relevant for the sake of a visual comparison of the tomographic results. Furthermore, the elevation information in Lupi et al. (2016) ERT is not correct because it is upward shifted of about 30 m. Generally, the two resistivity distributions can be considered consistent in the limits of their different spatial resolution, of their investigation depth, and of the time span intercurred between the two datasets acquisition (November 2012 for Lupi et
al., 2016, dataset, February 2022 for the dataset here presented). The presence of the "conductive domes" in Lupi et al., (2016) section, marked with the capital letter "D1-3" in figure 4, is here confirmed as well as the existence of more resistive areas between them. The main differences are related to the conductive areas placed respectively a) in the shallow portion (20 m from the surface) of the resistivity image between mud volcanoes 1 and 2 (marked with the capital letter "C1") and b) in the elevation range 150-180 m a.s.l. between volcanoes 2 and 3 (marked with the capital letter "C2"). The first one (C1) may be
an artifact of the inversion procedure and not a real feature of the subsoil. As already discussed in presenting figures 2 and 3, data involved in the inversion procedure that led to the conductive anomalies are those characterized by the largest experimental errors, both reciprocal and stacking. The same it is not true for the second conductive anomaly (C2) which can be considered representative of a real condition of the subsoil that was probably not existing in 2012 when Lupi et al., (2016) surveys were performed.



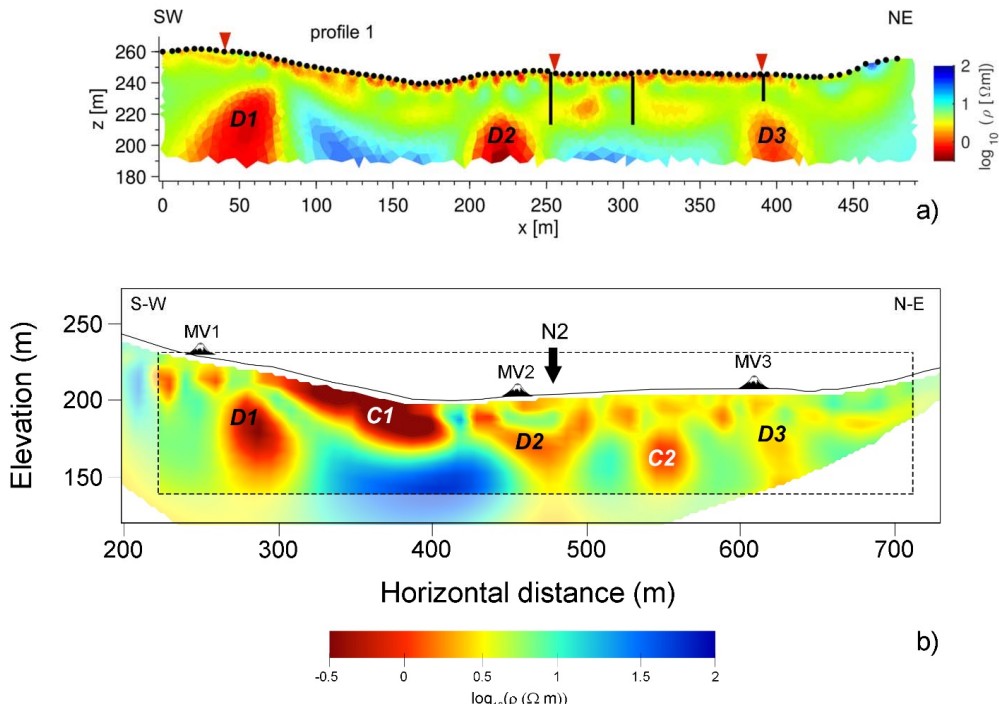

**Figure 4: comparison between Lupi et al. (2016) resistivity model (panel a) and the corresponding portion of the ERT_N1 DD resistivity model (black dotted line in panel b). The elevation information as well as mud volcano positions slightly differ in the two models. The meaning of the capital letters D1-3 and C1-2 is described in the text.**

### 3.2.1.2 Wenner-Schlumberger and Pole-Dipole data inversion.

As previously specified, for these datasets only stacking errors are available. The integration of the stacking errors in the inversion procedure, the lower noise level which usually characterize these geoelectrical array data, and the comparison with the DD inversion results are deemed to be sufficient to evaluate the reliability of the resistivity models.

As for DD data, the inversion procedure was preceded by the analysis of the contact resistances (figure 5, panels a and c) and of the stacking error spatial distributions (figure 5, panels b and d). The WS dataset is characterized (figure 5 a and c) by low contact resistances and low stacking error values almost uniformly distributed along the profile and in depth (differently from what observed for the DD dataset). The P-DD dataset had an electrode with a bad contact resistance (as highlighted by the two "red bands" in fig. 5 panels b and d). The P-DD data were acquired soon after the WS data which show no evidence of bad



resistivity contact. Thus, the observed worsening of the contact resistance is probably due to a tourist in the area who unplugged
one of the electrodes. The stacking errors amplitude distribution agrees with the measuring conditions. Negative data were
also removed from the maps in figure 5.

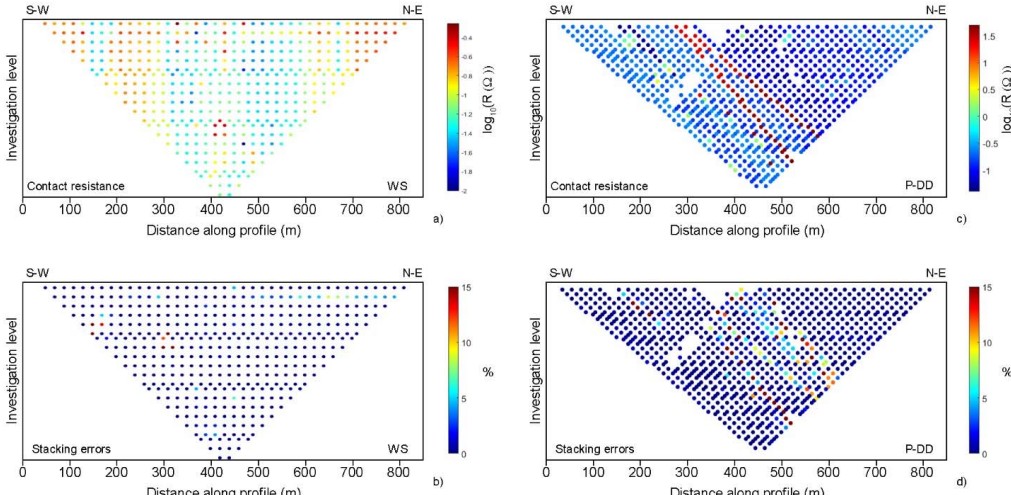

**Figure 5: Panel a): distribution of the contact resistances related to the WS dataset. Panel b): distribution of the contact resistances**
**related to the P-DD dataset. Panel c): stacking errors distribution of the WS dataset. Panel d): stacking errors distribution of the P-**
**DD dataset. The contact resistances and error magnitude can be read on the colour scale placed on the right side of each panel. Note**
**that, for the contact resistances, the limits of WS and P-DD colour scales are different.**

The datasets were then filtered by eliminating data associated with a percentage stacking error higher than 5% and inverted
with the same strategy adopted for the DD dataset. Inversion results are shown in figure 6 along with the DD model previously
shown in figure 3 (upper panel).

Considering the differences in terms of sensitivity and vertical/horizontal resolution of the three adopted electrodic
configurations, the resistivity models (figure 6) show a general agreement both in terms of resistivity contrast and spatial
distribution of the electrical properties in the subsoil with an investigation depth which, in the P-DD model, goes down to -50
m b.s.l.. DD and P-DD are similar except for the area between 300 and 400 meters (horizontal distance) (magenta dashed
rectangle in figure 6). Here, the conductive surficial layer in the DD ERT is replaced both in the P-DD and in the WS by a less
extended and less conductive area whereas the deeper resistive area visible in the DD is divided in two by a conductive anomaly
(more continuous in the P-DD image). WS, as emerged by the analysis of the stacking error distribution, is the only dataset
which does not show any error value increase in this area. This condition ensures reliability of the resistivity distribution



imaged in this area by the WS model and, by similarity, also by the P-DD model. For this reason, the geological and structural

interpretation of the geoelectrical model will be performed on the P-DD one.









**Figure 6: resistivity models obtained by inverting the DD dataset (upper panel), the WS dataset (middle panel) and the P-DD dataset (lower panel). All models are presented by using the same colour scale. The position of the mud volcanoes (MV1, MV2 and MV3)**
**along the profile is marked on top of each section along with the indication (vertical black arrow) of the crossing point between the section ERT_N1 and the section ERT_N2. The magenta dashed rectangles indicate the area of higher dissimilitude between the tomographic results. The dotted and dashed lines reported on the P-DD section represent the subsurface area investigated by the WS and DD sections, respectively.**


### 3.2.2 ERT_N2 profile

For the ERT_N2 profile, it was adopted the same approach used for ERT_N1 profile except for the comparison with previous/independent geophysical data. In Lupi et al. (2016), in fact, several profiles oriented as ERT_N2 are presented but they have a higher spatial resolution and lower investigation depth with respect to ours, to be used as a reference point for a
fruitful comparison. For the DD dataset, the mean percentage reciprocal error is 8.3% while the mean percentage stacking error is 4.4%. By integrating the two different error types in the inversion procedure, similar results, in terms of resistivity distributions, came out (not shown here). The reliability of the resistivity models is also ensured by the similarities between DD models and WS and P-DD ones (figure 7). All models show the presence of strong resistivity contrasts from 300 m (horizontal distance) till the end of the profile. In all models and in their central portion, a uniformly high resistive area is
visible which, as it emerges from the DD and from the P-DD sections, is confined in depth down to ~100 m a.s.l.. As for the ERT_N1 profile, DD and P-DD are more sensitive to horizontal resistivity variations and, hence, depict a more detailed resistivity distribution.



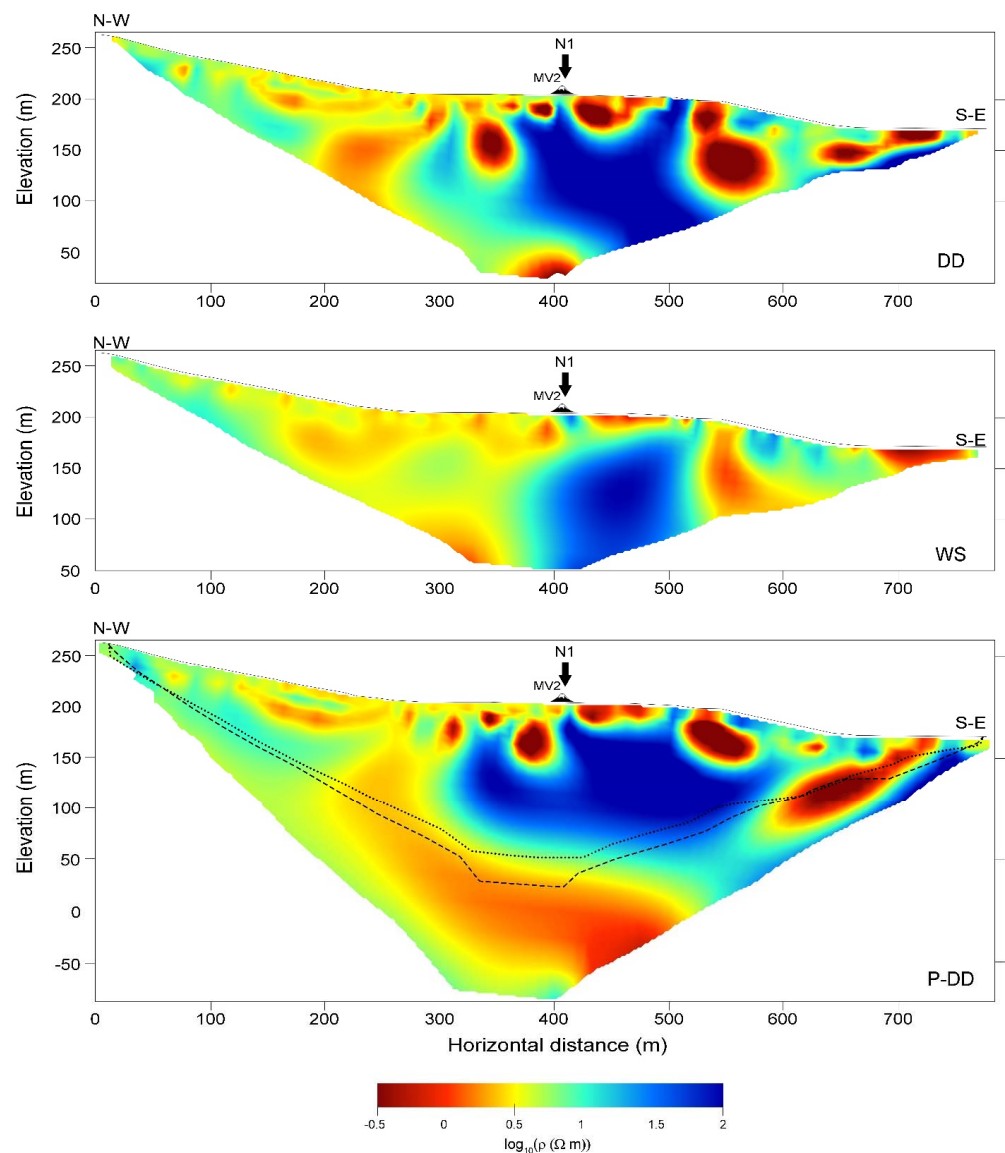



**Figure 7: resistivity models obtained by inverting the DD dataset (upper panel), the WS dataset (middle panel) and the P-DD dataset (lower panel). All models are presented by using the same colour scale. The position of the mud volcano MV1, placed off profile at about 20 m from the ERT_N2 trace, is marked on top of each section along with the indication (vertical black arrow) of the crossing point between the section N2 and the section N1. The dotted and dashed lines reported on the P-DD section represent the subsurface area investigated by the WS and DD sections, respectively.**

**3.2.3 3D visualization of the ERT profiles**

The spatial continuity of the two tomographic results and a clearer image of the resistivity distribution within the subsoil can be obtained by simultaneously plotting ERT_N1 and ERT_N2 (fig. 8). This operation shows a good match between N1 and N2 with a quite good consistency between the two resistivity distributions in the common investigated portion of subsoil.

Figure 8 also reveals how the Northern sector of the Nirano Salse area strongly differs, in term of resistivity distribution, from

the southern one. To the North, the subsoil is more homogeneous and generally conductive ($\rho < 10$ $\Omega$m). To the South, strong resistivity contrasts (both lateral and vertical) characterize the resistivity sections. Mud volcanoes are generally located over conductive anomalies.



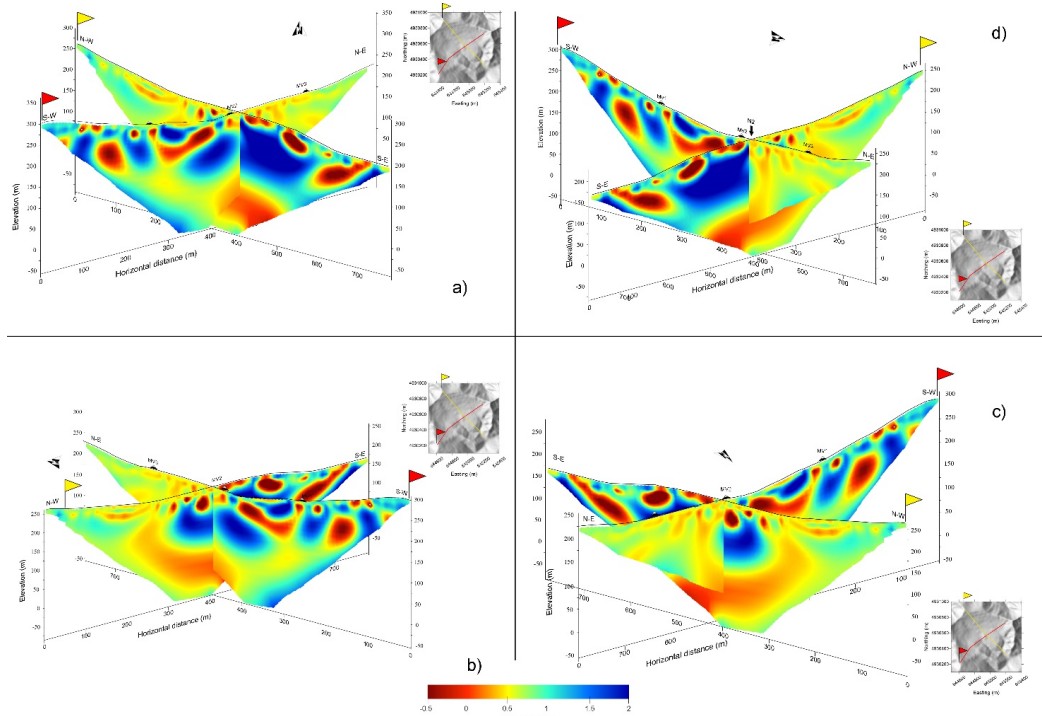

**Figure 8: 3D arrangements of the tomographic results. Starting from panel a) a counter clockwise rotation of the tomographic result**
**is performed. The yellow and red flags on top of each section indicate the starting point of the geoelectrical surveys. The inset in**
**each panel shows the location of the survey.**

## 4 Geological model

The interpretation of the two geoelectric pole dipole profiles (figures 9 and 10) presented in this study allows to shed some
light on the structural setting of the Nirano area and propose a novel geological model that can account for the position of the
vents, their activity, and the morphology of the area.

The resistivity features recovered by the presented models will be discussed in terms of i) geological boundaries, ii) mechanical
conditions (in terms of fracture presences), iii) fluid presence, and iv) fluid compositions.

The Miocene marly rocks and the Argille Azzurre (Pliocene FAA), the most relevant geological units expected in the area, are
characterized by a different electrical fingerprint with the former generally more resistive than the latter. Fluid presence can



affect the measured resistivity in a more complex way. On a first approximation, a gas phase is expected to enhance resistivity whereas a liquid phase electrical signature depends on the liquid type; saline fluids lower the resistivity whereas hydrocarbons, for example, have the opposite effect.

### 4.1 SW-NE ERT1 section

The SW-NE ERT1 section is characterized by the presence of a shallow conductive layer (2-10 $\Omega$m) that is associated to the Argille Azzurre outcrop (FAA). The SW portion of the model shows strong resistivity contrasts (ranging from 0.3 to 80 $\Omega$m) down to about 50 m of elevation a.s.l. (from 0 to about 450 m of distance along profile); the alternating resistivity anomalies often assume a subvertical elongated shape. In contrast, the NE portion of the model is mostly conductive (ranging from 2 to 20 $\Omega$m) with a layered structure. It is worth noting that the passage between these two different resistivity distribution domains

occurs in correspondence of the axis of a monocline or of a fault (horizontal distance about 500 m, grey shaded rectangle in figure 9 labelled as "F") that may be related at depth to the structural trap for fluid accumulation (Bonini, 2007). We interpret the subvertical resistivity contrast present towards SW as faults that spatially well corresponds at faults inferred also by other authors (Bonini, 2008) (red arrows in figure 9). These faults slightly offset the base of the FAA represented with the thick dashed red line (Fig. 9). The conductive layered structure toward NE is interpreted as the presence of multiple aquifers (low

resistivity areas, indicated in figure 9 by blue dashed lines), that well agree with the presence of sandy turbidite deposits in the FAA ( Castaldini et al., 2017; Giambastiani et al.; 2022) and are locally interconnected between vents MV2 e MV3. A shallow sub-horizontal connection also exists between vents MV1 and MV2 (labelled as "C" in figure 9), although characterized by lower resistivity values. These very low resistivity values (as low as 0.3 $\Omega$m) nearby MV1 and MV2, were also imaged by the shallow ERT model presented by Lupi et al. (2016) and were interpreted to reflect the presence of high-salinity fluids. This

interpretation, however, is not supported by recent fluid conductivity measurements performed directly in situ at some vents (Giambastiani et al., 2022; Oppo, 2011; Oppo et al., 2013) with conductivity values ranging between 10 to 21 mS/cm (that correspond to a resistivity value ranging 0.5 to 1 $\Omega$m) typical of brackish water. To justify the observed low resistivity values at depth, it is hence required to add at least another conductivity phase (Glover & Hole, 2000). We explore two possibilities. The first one is clay content and surface conductivity phase. In saturated soil, electrical current flow by means of ions that are

present in the saturating fluid (brine). If the grain size is very small, such as in clay, electrical current will easily flow through the pore fluid making it less resistive. Particle size distributions of mud samples collected in some of the gryphons (Giambastiani et al., 2022) support this the extra conduction phase as they indicate the presence of about 35% of clay, 56% of silt, and remanent of sand. The second possibility to be considered is the presence of iron sulphides (pyrite) due to precipitation of hydrogen sulphide ($H_2S$). Supporting this condition is the low amount of $H_2S$ at surface both among the emitted gas and

solutions since it can precipitate as iron sulphides due to the largely available iron in the siliciclastic sedimentary units (Oppo, 2011; Oppo et al., 2013). Thus, we can interpret the very low resistivity anomaly "C" as very shallow reservoir hosting brackish water and where one of the above-mentioned conductivity phases is acting. The subvertical low (NE) ("F" in figure 9) and very low resistivity (SW) ("C" in figure 9) anomalies, here interpreted as favourable routes for fluid uprising, converge deeply



(above 50 m elevation) towards a common deep conductor. This common conductive zone may represent a the Cigarello Fm

clay unit (Gasperi et al., 2005).

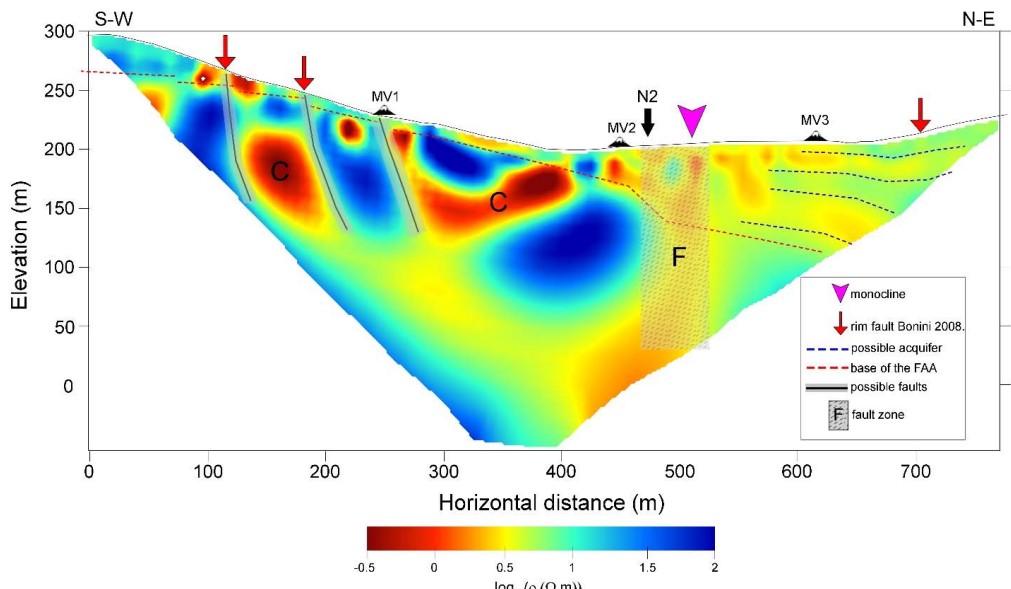

**Figure 9: interpreted ERT_N1 profile. The capital letter "C" shows the location of the conductive anomalies associated to favourable routes for fluids uprising. The capital letter "F" indicates the area probably associated to a fault zone.**

**4.2 NW-SE ERT2**

The NW-SE ERT2 section (fig. 10) is almost parallel to the axis of the anticline. In this section, we also observe a relevant heterogeneity in resistivity distribution. As for the NE-SW ERT1, the NW portion of the section is regular and shows the presence of a sub-horizontal stratification, which may indicate the presence of several aquifers (blue dashed lines in fig. 10). The SE portion of the model shows more enhanced resistivity contrasts as the SW portion of the of ERT1 model, but with a

less complex pattern. The transition between the more regular NW portion and the more heterogeneous SE is spatially coincident with the presence of the fault zone (two central red arrows in fig. 10) also described in ERT1, which forms a small angle with this section. Indeed, down to 50 m of elevation, local shallow conductive zones are hosted in a resistive layer that is sharply interrupted in the S-Easternmost area, at about 650 m along profile, where we infer the presence of a fault (blue arrow) whose surface intersection is located near a dry gas seep (yellow arrow in fig. 10). This fault may be related a Ne-SW

trending faults that are strike-slip transfer structures reported by Gasperi et al.,(2005) in this area (blue arrow in fig. 10). Thus,



we interpret the resistive layer as the Termina marls and Pantano Fm (Miocene) unit hosting local conductive anomalies associable to very shallow fluid reservoirs. In the deeper part of the model, along the whole profile, a deep conductor gently dipping towards SE is imaged. This conductive zone spatially well coincides with the one recovered in ERT1, and thus is interpreted as the Cigarello Fm claystone (Gasperi et al., 2005).

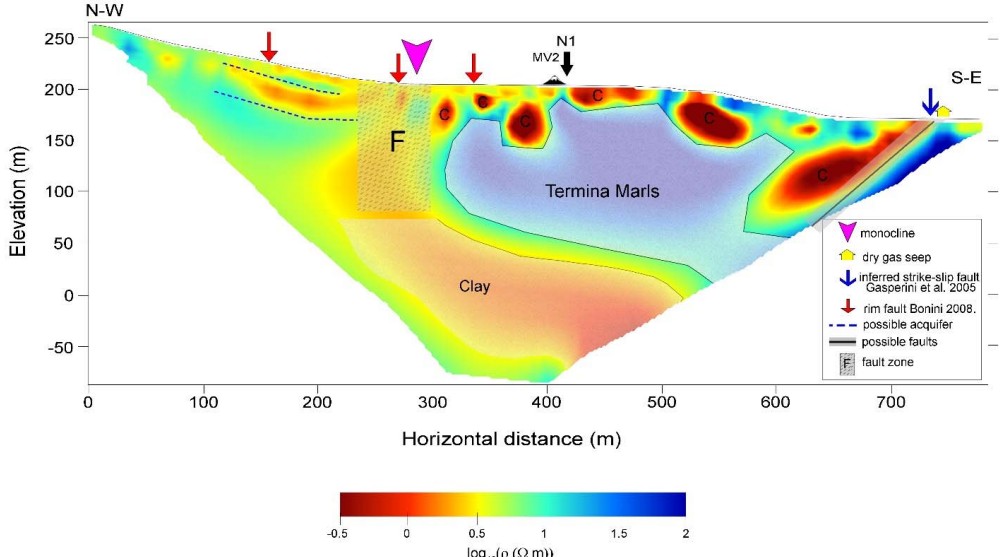


**Figure 10: interpreted ERT_N2 profile. The capital letter "C" shows the location of the conductive anomalies associated to favourable routs for fluids uprising.**

Summing up, the resistivity features recovered by the two models well match in the intersection area. Interestingly, an undisturbed layered system characterizes the norther sector (NW and NE) of the Nirano area in contrast to a more heterogeneous (SE) and even chaotic (SW) resistivity distribution in the southern sector. These major heterogeneities are located just where also the "bowl" morphology is irregular: in the SE corner a small creek cuts across the rim and two relatively NW-SE oriented steep ridges border the southwestern flank of the circular structure. It could be speculated that regional NW-SE and NE-SW oriented faults interacting and interfering with each other contribute to the formation of the geomorphological bowl. The tomographic images, in fact, suggest the presence of linear faults and fracture zones and not of circular structures as previously speculated by Bonini (2011) and Sciarra et al. (2019). These are important observations, because they do not support a gravitational collapse of the "bowl" area as previously assumed. On the other hand, the deepest parts of both models (below 70 m elevation) share a common conductive feature, associated with a clayey horizon (Cigarello Fm) that may act as



an impermeable horizon for deep uprising fluids. These deep fluids can reach the surface only where the fault zone break this
sealing unit allowing for gas and mud venting at the surface.

## 5 Conclusions

The geophysical investigation performed in the Nirano Salse provided new information on their deep structure. The pole-
dipole geoelectric sections showed the existence of different structural conditions in the bowl-shaped morphology of the area.
The Northern sector of Nirano subsurface seems to be characterized by a regular vertical stratification which can be associated
to the presence of multilayer aquifers within the Argille Azzurre Fm. The Southern sector, on the contrary, is dominated by
the presence of subvertical discontinuities which can be associated with faults and fracture zones. In the geolectrical sections,
conductive zones can be associated to fluid accumulation areas. Fluid presence, however, is not sufficient to justify the
extremely low resistivity values observed in both sections. We suggest that the low resistivity values may be associated with
i) clay content and surface conductivity phase and/or ii) presence of iron sulphides (pyrite) due to precipitation of hydrogen
sulphide.
Regardless of what the explanation for the observed resistivity values is, the areas containing fluids (gas and water) are in the
shallow portion (50-100 m) of the subsoil and in correspondence with the main mud volcanoes. It is therefore reasonable to
assume that these fluid accumulation zones constitute the surficial reservoirs of volcanoes. Larger scale of investigation
geophysical survey could disclose the existence of a deeper fluid storage and its relationship with these shallow, and localized
reservoirs. The shallow fluid reservoirs have a relatively small volume, and they are sparse in the sections clearly showing that
there is not a shallow fluid caldera below the Nirano Salse. This observation has important implications for the safety of the
site. The most hazardous area, in fact, are those closer to the mud vents where the shallow small reservoirs are located and not
the whole area of the "bowl". This could help pinpoint the routes for tourists' access and fence out areas of potential
liquefaction and mud eruption. This methodology for assessing local hazard could be extended to other mud volcanoes areas
around the world.
The geoelectrical sections also highlighted the presence of faults, which are linear features oriented as most of the structures
in this sector of the Apennines (NW-SE) and not circular collapse rim faults as previously thought. Likely, they provide
gas/fluid flow routes from deep sources  to the shallow reservoirs and finally to the mud vulcanoes. A relevant example can
be the major fault zone in the central portion of ERT_N1, which could be thought as the main pathway for the fluids and gas
to reach the surface from their deep reservoirs (located out of ERT investigation depth limit). This major fault zone also offsets
down-to-the-NE the Argille Azzurre Fm (FAA) and it accommodates deformation along the monocline flexure, which is well
exposed north of the Nirano Salse.

Acknowledgments



The authors thanks Luciano Callegari, Marzia Conventi and Maria Morena for their support in the logistics and the warm welcome. This study was performed with the support from the Fiorano Modenese Municipality (Italy) and the Management Authority for Park and Biodiversity of Central Emilia (Ente di Gestione per i Parchi e la Biodiversità Emilia Centrale, Italy).

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
