# Peer review of "Fluid Conduits and Shallow Reservoir Structure Defined by Geoelectrical Tomography at the Nirano Salse (Italy)"

_EGUsphere, 2023_

## Author Response (AR1)

**In this document there is a point-to-point response (highlighted in green) to the reviews including a list of all relevant changes made in the manuscript (highlighted in cyan).**

**Referee 1**

**General comments**

This manuscript describes the application of 2D electrical resistivity surveys for the deeper characterization of an area where several active mud volcanoes are presents (Nirano Salse, Northern Apennines, Italy). The main goal is to individuate fluid reservoirs as well as their connection among aquifers, conduits, and mud reservoirs. This information is fundamental for the understanding of Mud volcanoes dynamics and the assessment of local site hazards for a safer tourist access to the area. Finally, the methodology is potentially applicable to other mud volcanoes areas around the world.

In general, the paper is clear and well written, however some paragraphs could be improved.

**Specific comments**

- Section **2 Geological introduction to the area** and section **3.1 Previous geophysical data**.

I suggest the authors add a figure summarizing geological information and previous geophysical investigations.

Following the Referee suggestion, we modified Fig. 1 adding the required information.

- Section **3.2.1.1 Dipole-Dipole data inversion**

Line 233: 40% of reciprocal error is a quite high threshold. Please, explain how the authors Please, explain how the authors established as effective this threshold and why they didn't consider lower values of reciprocal error for data filtering.

The Referee is right: 40% of reciprocal error is a quite high threshold. In fact, it was used only for a preliminary filtering stage. As written in the text, an additional following filtering stage was based on the observation of resistivity data as a profile plot. Data characterized by abrupt resistivity variations were discarded.

We modified the text in order clearly explain the whole data preprocessing.

Line 237: respect the reciprocal error filtering, how many data were filtered using the stacking error method? what is the percentage of stacking error considered?

This information was missing in the text and was added in the reviewed version. We thank the Referee for his/her useful comment.

- Section **3.2.1.2 Wenner-Schlumberger and Pole-Dipole data inversion**

How many data were filtered for WS and PD surveys using the stacking error method?

The required information was added in the main text

- Section **4.1 SW-NE ERT1 section**

Lines 399-400: the "common deep conductor" needs to be better highlighted in figure 9.

Figure 9 was modified according to the referee comment.

**Technical corrections**

- Line 106: "deep **electric geoelectric** tomography surveys". There is an unnecessary repetition.

- The text in figure 1 is too small.

- Line 259: change "resistivity" with "Resistivity".

- Figure 5: check the order of the panels as described in the caption.

- Figure 8: the quality of the image must be improved.

- Line 414: change "Ne-SW" with "NE-SW".

We made all the requested technical corrections in the text.

**Referee 2**

Major comments:

- Did the authors try the inversions with amplified stacking errors for P-DD and WS data? If so, are there any effects of large error in the model? Since the authors mention that stacking error tends to be an underestimated measurement error compared to reciprocal error in Line 229 – 231, I am concerned about the effect of the small measurement error in P-DD inversion results used in interpretation. Furthermore, the authors have both reciprocal and stacking error for DD data in N1, it is easy to estimate the factor of amplification for stacking error, which the authors mention in Line 231. For this version of manuscript, some comments seem necessary for the reason why the authors use the stacking errors as they are in WS and P-DD inversions.

Thank you to the Referee for his/her first comment.

For N1 profile (DD dataset), the ratio between the median reciprocal errors and the median stacking error is 2.4. Thus, we multiplied the stacking error for the P-DD and WS dataset for 2.4 and repeated the inversion procedure.

WS dataset:

Keeping unchanged the filtering procedure ("The datasets were then filtered by eliminating data associated with a percentage stacking error higher than 5%") and considering the staking error distribution (see figure 5 of the manuscript), the inverted WS dataset (with amplified errors) was composed by the same measuring points of the one used to obtain the model of figure 6 (central panel). Thus, no additional data were filtered out. The inversion produced a "new" WS resistivity model that has lower rms compared to the model shown in figure 6. However no relevant changes are observable.

P-DD dataset:

As regard the P-DD datasets, by amplifying the stacking errors and keeping unchanged the filtering rules, a larger number of data was filtered out especially in the central and shallow part of the section. The resulted resistivity model is hence almost equal to the one shown in figure 6 (lower panel) except for the shallow part between 300m and 400m along the profile. In this area, the conductor "C" is more pushed toward the surface. Also in this case, like for the WS, the geological interpretation of the model does not significantly vary.

We can hence conclude that, at least in this case, the stacking errors amplification does not modify in a relevant way the inversion results provided that the inverted dataset is unchanged.

If the staking error amplification produces, like happened for the P-DD dataset, a dataset reduction (due to the adoption of filtering rules), modifications are observed. In this case, the more data are discarded the greater the inversion result difference will be.

We will add these observations in the main text (section 3.2.1.2).

- What is robustly inferred in the S-W portion of the P-DD model? Although the authors adopt P-DD model for its similarity to WS model in the rectangular area with magenta dashed lines (Fig. 6), I was surprised to find that the resistive/conductive patterns in the S-W portion, at distance < 300m, are almost reversed between WS and P-DD models. For example, in Fig. 9, the left "C"-labelled conductor in P-DD model corresponds to a resistive anomaly in WS model in Fig. 6, at the distance of 180m along the profile and at the elevation of 200m. In the section 4.2, the authors discussed the reason of conductivity value of 0.3 Ohm.m of the left "C"-labeled conductor in Fig. 9 and draw lines for possible faults nearby the C-labeled conductor. At present, it seems hard to believe the SW portion of the P-DD model like the authors due to the large inconsistency with the WS model. Unless the authors explain the robustness of the S-W portion of P-DD model compared to WS model, readers cannot follow the authors' discussion about the S-W portion of P-DD model along N1.

As regard the second Referee comment, the P-DD robustness model was evaluated by the comparison with the DD ones since this last was the only with a reciprocal error available and it is really sensitive to lateral resistivity contrast.

Indeed, the P-DD array has the advantage to mediate between the WS and DD arrays characteristics.

The P-DD and the DD models are similar with the exception of the shallow central part of the model (magenta dashed rectangle in fig. 6). Only in this area, we used the comparison with the WS because, as explained in the text, the DD results area less constrained by the data due to data quality and filtering operations.

In any case, the differences between the WS and the P-DD model for distances along profile < 300m can be ascribed to:

- the different sensitivity of the arrays to lateral resistivity variation (much higher in the P-DD than in the WS) and the substantial difference in data spatial coverage (P-DD has a higher data coverage than WS, see figure 5). This, for example, may explain what imaged by the two datasets just below MV1 (figure 6). Here, the WS model depicts the presence of a unique conductive body while P-DD (but also DD with reciprocal errors) shows the presence of two smaller conductive nucleus.

- Different investigation depth. The P-DD has an investigation depth which is almost two times the one of the WS. Thus, resistivity anomalies recovered at the bottom of the WS model are not constrained by data sensing the deeper portion. For example,

the resistive nucleus depicted in the WS at about 180m along profile extends up to the lower limit of the sections. In the P-DD model, the same resistivity anomaly is confined at shallower depth and overlays a conductive area (not imaged by the WS due to the lack of deeper data).

We will add this discussion in the main text  (section 3.2.1.2).

Minor comments:

Figure 1b

This panel is not very informative about topography of the area. Can the authors modify this panel so that readers easily recognize where is high and low elevations. Furthermore, if it doesn't makes the panel too crowded, the reviewer wants to see the survey line of Liu et al. (2016) in this panel only for the line overlapping ERT-N1.

Figure 1 was modified according to the referee comment.

Line 337

Please mention that the P-DD model along N2 will be used for interpretation like N1 profile within this section.

Done

Line 370 - 400

Section 4.1 consists of only one paragraph and this paragraph is too long to read through. Can the authors kindly split it to several paragraphs corresponding to different subjects of discussion?

 Done

Line 370

Please specify, in this sentence, which figure readers should look at to follow the authors' discussion. It's hard to find "a shallow conductive layer" without any guidance.

 Done

Line 381

MV2 e MV3 -> MV2 and MV3 ?

Done

Line 335 and Line 407 "as for"

I think "as for" is usually used similar to "about". The authors' use of "as for " similar to "like" is common? This use was confusing to me.

We modified the main text.

Figure 5

What do lacks of dots mean in the P-DD panels for the contact resistivity and stacking error? I cannot find any explanation about the lacks of dots, although the red bands in c) panel is well explained.

Lack of dots is due to the presence of negative data which were filtered out from the dataset used for creating the figure. We added this information in the figure caption.

Figure 6

Please mention which profile the models are shown for in the caption. This is also the case for Figure 7, where no clear mention for the N2 profile in the caption.

Done